# Bayesian Alignments of Warped Multi-Output Gaussian Processes

**Markus Kaiser**
Siemens AG
Technical University of Munich
markus.kaiser@siemens.com

**Clemens Otte**
Siemens AG
clemens.otte@siemens.com

**Thomas Runkler**
Siemens AG
Technical University of Munich
thomas.runkler@siemens.com

**Carl Henrik Ek**
University of Bristol
carlhenrik.ek@bristol.ac.uk

## Abstract

We propose a novel Bayesian approach to modelling nonlinear alignments of time series based on latent shared information. We apply the method to the real-world problem of finding common structure in the sensor data of wind turbines introduced by the underlying latent and turbulent wind field. The proposed model allows for both arbitrary alignments of the inputs and non-parametric output warpings to transform the observations. This gives rise to multiple deep Gaussian process models connected via latent generating processes. We present an efficient variational approximation based on nested variational compression and show how the model can be used to extract shared information between dependent time series, recovering an interpretable functional decomposition of the learning problem. We show results for an artificial data set and real-world data of two wind turbines.

## 1 Introduction

Many real-world systems are inherently hierarchical and connected. Ideally, a machine learning method should model and recognize such dependencies. Take wind power production, which is one of the major providers for renewable energy today, as an example: To optimize the efficiency of a wind turbine the speed and pitch have to be controlled according to the local wind conditions (speed and direction). In a wind farm turbines are typically equipped with sensors for wind speed and direction. The goal is to use these sensor data to produce accurate estimates and forecasts of the wind conditions at every turbine in the farm. For the ideal case of a homogeneous and very slowly changing wind field, the wind conditions at each geometrical position in a wind farm can be estimated using the propagation times (time warps) computed from geometry, wind speed, and direction [21, 4, 18]. In the real world, however, wind fields are not homogeneous, exhibit global and local turbulences, and interfere with the turbines and the terrain inside and outside the farm and further, sensor faults may lead to data loss. This makes it extremely difficult to construct accurate analytical models of wind propagation in a farm. Also, standard approaches for extracting such information from data, e.g. generalized time warping [24], fail at this task because they rely on a high signal to noise ratio. Instead, we want to construct Bayesian nonlinear dynamic data based models for wind conditions and warpings which handle the stochastic nature of the system in a principled manner.

In this paper, we look at a generalization of this type of problem and propose a novel Bayesian approach to finding nonlinear alignments of time series based on latent shared information. We view the power production of different wind turbines as the outputs of a multi-output Gaussian process

(MO-GP) [1] which models the latent wind fronts. We embed this model in a hierarchy, adding a layer of non-linear alignments on top and a layer of non-linear warpings [19, 14] below which increases flexibility and encodes the original generative process. We show how the resulting model can be interpreted as a group of deep Gaussian processes with the added benefit of covariances between different outputs. The imposed structure is used to formulate prior knowledge in a principled manner, restrict the representational power to physically plausible models and recover the desired latent wind fronts and relative alignments. The presented model can be interpreted as a group of $D$ deep GPs all of which share one layer which is a MO-GP. This MO-GP acts as an interface to share information between the different GPs which are otherwise conditionally independent.

The paper has the following contributions: In Section 2, we propose a hierarchical, warped and aligned multi-output Gaussian process (AMO-GP). In Section 3, we present an efficient learning scheme via an approximation to the marginal likelihood which allows us to fully exploit the regularization provided by our structure, yielding highly interpretable results. We show these properties for an artificial data set and for real-world data of two wind turbines in Section 4.

## 2 Model Definition

We are interested in formulating shared priors over a set of functions $\{f_d\}_{d=1}^{D}$ using GPs, thereby directly parameterizing their interdependencies. In a traditional GP setting, multiple outputs are considered conditionally independent given the inputs, which significantly reduces the computational cost but also prevents the utilization of shared information. Such interdependencies can be formulated via *convolution processes (CPs)* as proposed by Boyle and Frean [5], a generalization of the *linear model of coregionalization (LMC)* [13, 7]. In the CP framework, the output functions are the result of a convolution of the latent processes $w_r$ with smoothing kernel functions $T_{d,r}$ for each output $f_d$, defined as

$$f_d(\boldsymbol{x}) = \sum_{r=1}^{R} \int T_{d,r}(\boldsymbol{x} - \boldsymbol{z}) \cdot w_r(\boldsymbol{z}) \, d\boldsymbol{z}. \tag{1}$$

In this model, the convolutions of the latent processes generating the different outputs are all performed around the same point $\boldsymbol{x}$. We generalize this by allowing different *alignments* of the observations which depend on the position in the input space. This allows us to model the changing relative interaction times for the different latent wind fronts as described in the introduction. We also assume that the dependent functions $f_d$ are latent themselves and the data we observe is generated via independent noisy nonlinear transformations of their values. Every function $f_d$ is augmented with an alignment function $a_d$ and a warping $g_d$ on which we place independent GP priors.

For simplicity, we assume that the outputs are evaluated all at the same positions $\boldsymbol{X} = \{\boldsymbol{x_n}\}_{n=1}^{N}$. This can easily be generalized to different input sets for every output. In our application, the $\boldsymbol{x_n}$ are one-dimensional time indices. However, since the model can be generalized to multi-dimensional inputs, we do not restrict ourselves to the one-dimensional case. We note that in the multi-dimensional case, reasoning about priors on alignments can be challenging. We call the observations associated with the $d$-th function $\boldsymbol{y_d}$ and use the stacked vector $\boldsymbol{y} = (\boldsymbol{y_1}, \ldots, \boldsymbol{y_D})$ to collect the data of all outputs. The final model is then given by

$$\boldsymbol{y_d} = g_d(f_d(a_d(\boldsymbol{X}))) + \boldsymbol{\epsilon_d}, \tag{2}$$

where $\boldsymbol{\epsilon_d} \sim \mathcal{N}(0, \sigma_{y,d}^2 \mathbf{I})$ is a noise term. The functions are applied element-wise. This encodes the generative process described above: For every turbine $\boldsymbol{y_d}$, observations at positions $\boldsymbol{X}$ are generated by first aligning to the latent wind fronts using $a_d$, applying the front in $f_d$, imposing turbine-specific components $g_d$ and adding noise in $\boldsymbol{\epsilon_d}$.

We assume independence between $a_d$ and $g_d$ across outputs and apply GP priors of the form $a_d \sim \mathcal{GP}(\mathrm{id}, k_{a,d})$ and $g_d \sim \mathcal{GP}(\mathrm{id}, k_{g,d})$. By setting the prior mean to the identity function $\mathrm{id}(x) = x$, the standard CP model is our default assumption. During learning, the model can choose the different $a_d$ and $g_d$ in a way to reveal the independent shared latent processes $\{w_r\}_{r=1}^{R}$ on which we also place GP priors $w_r \sim \mathcal{GP}(0, k_{u,r})$. Similar to Boyle and Frean [5], we assume the latent processes to be independent white noise processes by setting $\mathrm{cov}[w_r(\boldsymbol{z}), w_{r'}(\boldsymbol{z'})] = \delta_{rr'}\delta_{\boldsymbol{zz'}}$. Under this prior, the $f_d$ are also GPs with zero mean and $\mathrm{cov}[f_d(\boldsymbol{x}), f_{d'}(\boldsymbol{x'})] = \sum_{r=1}^{R} \int T_{d,r}(\boldsymbol{x} - \boldsymbol{z}) T_{d',r}(\boldsymbol{x'} - \boldsymbol{z}) \, d\boldsymbol{z}$.

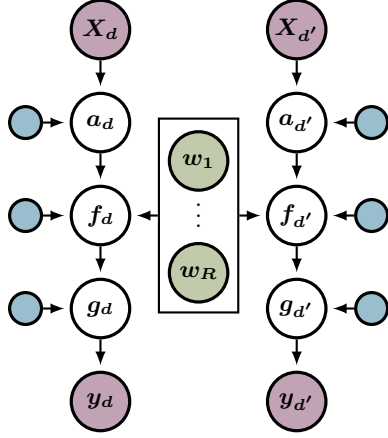

**Figure 1:** The graphical model of AMO-GP with variational parameters (blue). A CP, informed by $R$ latent processes, models shared information between multiple data sets with nonlinear alignments and warpings. This CP connects multiple deep GPs through a shared layer.

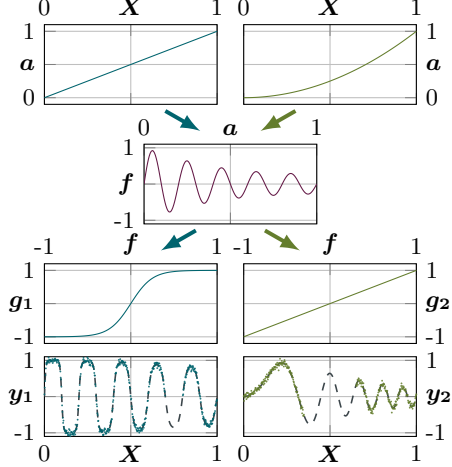

**Figure 2:** An artificial example of hierarchical composite data with multiple observations of shared latent information. This hierarchy generates two data sets using a dampened sine function which is never observed directly.

Using the squared exponential kernel for all $T_{d,r}$, the integral can be shown to have a closed form solution. With $\{\sigma_{d,r}, \boldsymbol{\ell_{d,r}}\}$ denoting the kernel hyper parameters associated with $T_{d,r}$, it is given by

$$\mathrm{cov}[f_d(\boldsymbol{x}), f_{d'}(\boldsymbol{x'})] = \sum_{r=1}^{R} \frac{(2\pi)^{\frac{K}{2}} \sigma_{d,r} \sigma_{d',r}}{\prod_{k=1}^{K} \hat{\ell}_{d,d',r,k}^{-1}} \exp\left( -\frac{1}{2} \sum_{k=1}^{K} \frac{(x_k - x'_k)^2}{\hat{\ell}_{d,d',r,k}^2} \right), \tag{3}$$

where $\boldsymbol{x}$ is $K$-dimensional and $\hat{\ell}_{d,d',r,k} = \sqrt{\ell_{d,r,k}^2 + \ell_{d',r,k}^2}$.

## 3 Variational Approximation

Since exact inference in this model is intractable, we present a variational approximation to the model's marginal likelihood in this section. A detailed derivation of the variational bound can be found in Appendix A. Analogously to $\boldsymbol{y}$, we denote the random vectors which contain the function values of the respective functions and outputs as $\boldsymbol{a}$ and $\boldsymbol{f}$. The joint probability distribution of the data can then be written as

$$\mathrm{p}(\boldsymbol{y}, \boldsymbol{f}, \boldsymbol{a} \,|\, \boldsymbol{X}) = \qquad\qquad \boldsymbol{a_d} \,|\, \boldsymbol{X} \sim \mathcal{N}(\boldsymbol{X}, \boldsymbol{K_{a,d}} + \sigma_{a,d}^2 \mathbf{I}),$$
$$\mathrm{p}(\boldsymbol{f} \,|\, \boldsymbol{a}) \prod_{d=1}^{D} \mathrm{p}(\boldsymbol{y_d} \,|\, \boldsymbol{f_d}) \, \mathrm{p}(\boldsymbol{a_d} \,|\, \boldsymbol{X}), \qquad \boldsymbol{f} \,|\, \boldsymbol{a} \sim \mathcal{N}(\boldsymbol{0}, \boldsymbol{K_f} + \sigma_f^2 \mathbf{I}), \tag{4}$$
$$\boldsymbol{y_d} \,|\, \boldsymbol{f_d} \sim \mathcal{N}(\boldsymbol{f_d}, \boldsymbol{K_{g,d}} + \sigma_{y,d}^2 \mathbf{I}).$$

Here, we use $\boldsymbol{K}$ to refer to the Gram matrices corresponding to the respective GPs. All but the convolution processes factorize over the different levels of the model as well as the different outputs.

### 3.1 Variational Lower Bound

To approximate a single deep GP, that is a single string of GPs stacked on top of each other, Hensman and Lawrence [11] proposed nested variational compression in which every GP in the hierarchy is handled independently. In order to arrive at their lower bound they make two variational approximations. First, they consider a variational approximation $\mathrm{q}(\hat{\boldsymbol{a}}, \boldsymbol{u}) = \mathrm{p}(\hat{\boldsymbol{a}} \,|\, \boldsymbol{u}) \, \mathrm{q}(\boldsymbol{u})$ to the true posterior of a single GP first introduced by Titsias [22]. In this approximation, the original model is augmented with *inducing variables* $\boldsymbol{u}$ together with their *inducing points* $\boldsymbol{Z}$ which are assumed to

be latent observations of the same function and are thus jointly Gaussian with the observed data. In contrast to [22], the distribution q($\boldsymbol{u}$) is not chosen optimally but optimized using the closed form q($\boldsymbol{u}$) $\sim \mathcal{N}(\boldsymbol{u} \,|\, \boldsymbol{m}, \boldsymbol{S})$. This gives rise to the Scalable Variational GP presented in [10]. Second, in order to apply this variational bound for the individual GPs recursively, uncertainties have to be propagated through subsequent layers and inter-layer cross-dependencies are avoided using another variational approximation. The variational lower bound for the AMO-GP is given by

$$
\begin{aligned}
\log \mathrm{p}(\boldsymbol{y} \,|\, \boldsymbol{X}, \boldsymbol{Z}, \boldsymbol{u}) \geq & \sum_{d=1}^{D} \log \mathcal{N}\Big(\boldsymbol{y_d} \,\Big|\, \boldsymbol{\Psi_{g,d}} \boldsymbol{K}_{\boldsymbol{u_{g,d}}\boldsymbol{u_{g,d}}}^{-1} \boldsymbol{m_{g,d}}, \sigma_{y,d}^2 \mathbf{I}\Big) - \sum_{d=1}^{D} \frac{1}{2\sigma_{a,d}^2} \mathrm{tr}(\boldsymbol{\Sigma_{a,d}}) \\
& - \frac{1}{2\sigma_f^2} \Big(\psi_f - \mathrm{tr}\Big(\boldsymbol{\Phi_f} \boldsymbol{K}_{\boldsymbol{u_f}\boldsymbol{u_f}}^{-1}\Big)\Big) - \sum_{d=1}^{D} \frac{1}{2\sigma_{y,d}^2} \Big(\psi_{g,d} - \mathrm{tr}\Big(\boldsymbol{\Phi_{g,d}} \boldsymbol{K}_{\boldsymbol{u_{g,d}}\boldsymbol{u_{g,d}}}^{-1}\Big)\Big) \\
& - \sum_{d=1}^{D} \mathrm{KL}(\mathrm{q}(\boldsymbol{u_{a,d}}) \,\|\, \mathrm{p}(\boldsymbol{u_{a,d}})) - \mathrm{KL}(\mathrm{q}(\boldsymbol{u_f}) \,\|\, \mathrm{p}(\boldsymbol{u_f})) - \sum_{d=1}^{D} \mathrm{KL}(\mathrm{q}(\boldsymbol{u_{y,d}}) \,\|\, \mathrm{p}(\boldsymbol{u_{y,d}})) \quad (5) \\
& - \frac{1}{2\sigma_f^2} \mathrm{tr}\left(\Big(\boldsymbol{\Phi_f} - \boldsymbol{\Psi_f^\top} \boldsymbol{\Psi_f}\Big) \boldsymbol{K}_{\boldsymbol{u_f}\boldsymbol{u_f}}^{-1} \Big(\boldsymbol{m_f} \boldsymbol{m_f^\top} + \boldsymbol{S_f}\Big) \boldsymbol{K}_{\boldsymbol{u_f}\boldsymbol{u_f}}^{-1}\right) \\
& - \sum_{d=1}^{D} \frac{1}{2\sigma_{y,d}^2} \mathrm{tr}\left(\Big(\boldsymbol{\Phi_{g,d}} - \boldsymbol{\Psi_{g,d}^\top} \boldsymbol{\Psi_{g,d}}\Big) \boldsymbol{K}_{\boldsymbol{u_{g,d}}\boldsymbol{u_{g,d}}}^{-1} \Big(\boldsymbol{m_{g,d}} \boldsymbol{m_{g,d}^\top} + \boldsymbol{S_{g,d}}\Big) \boldsymbol{K}_{\boldsymbol{u_{g,d}}\boldsymbol{u_{g,d}}}^{-1}\right),
\end{aligned}
$$

where KL denotes the KL-divergence. A detailed derivation can be found in Appendix A. The bound contains one Gaussian fit term per output dimension and a series of regularization terms for every GP in the hierarchy. The KL-divergences connect the variational approximations to the prior and the different trace terms regularize the variances of the different GPs (for a detailed discussion see [11]). This bound depends on the hyper parameters of the kernel and likelihood $\{\ell, \boldsymbol{\sigma}\}$ and the variational parameters $\{\boldsymbol{Z_{l,d}}, \boldsymbol{m_{l,d}}, \boldsymbol{S_{l,d}} \,|\, l \in \{\boldsymbol{a}, \boldsymbol{f}, \boldsymbol{d}\}, d \in [D]\}$.

The bound can be calculated in $\mathcal{O}(NM^2)$ time and factorizes along the data points which enables stochastic optimization. Since every of the $N$ data points is associated with one of the $D$ outputs, the computational cost of the model is independent of $D$. Information is only shared between the different outputs using the inducing points in $\boldsymbol{f}$. As the different outputs share a common function, increasing $D$ allows us to reduce the number of variational parameters per output, because the shared function can still be represented completely.

A central component of this bound are expectations over kernel matrices, the three $\Psi$-statistics $\psi_f = \mathbb{E}_{\mathrm{q}(\boldsymbol{a})}[\mathrm{tr}(\boldsymbol{K_{ff}})]$, $\boldsymbol{\Psi_f} = \mathbb{E}_{\mathrm{q}(\boldsymbol{a})}[\boldsymbol{K_{fu}}]$ and $\boldsymbol{\Phi_f} = \mathbb{E}_{\mathrm{q}(\boldsymbol{a})}[\boldsymbol{K_{uf}} \boldsymbol{K_{fu}}]$. Closed form solutions for these statistics depend on the choice of kernel and are known for specific kernels, such as linear or RBF kernels, for example shown in [8]. In the following subsection we will give closed form solutions for these statistics required in the shared CP-layer of our model.

## 3.2 Convolution Kernel Expectations

The uncertainty about the first layer is captured by the variational distribution of the latent alignments $\boldsymbol{a}$ given by q($\boldsymbol{a}$) $\sim \mathcal{N}(\boldsymbol{\mu_a}, \boldsymbol{\Sigma_a})$. Every aligned point in $\boldsymbol{a}$ corresponds to one output of $\boldsymbol{f}$ and ultimately to one of the $\boldsymbol{y_d}$. Since the closed form of the multi output kernel depends on the choice of outputs, we will use the notation $\hat{f}(\boldsymbol{a_n})$ to denote $f_d(\boldsymbol{a_n})$ such that $\boldsymbol{a_n}$ is associated with output $d$.

For simplicity, we only consider one single latent process $w_r$. Since the latent processes are independent, the results can easily be generalized to multiple processes. Then, $\psi_f$ is given by

$$
\psi_f = \mathbb{E}_{\mathrm{q}(\boldsymbol{a})}[\mathrm{tr}(\boldsymbol{K_{ff}})] = \sum_{n=1}^{N} \hat{\sigma}_{nn}^2. \quad (6)
$$

Similar to the notation $\hat{f}(\cdot)$, we use the notation $\hat{\sigma}_{nn'}$ to mean the variance term associated with the covariance function $\mathrm{cov}[\hat{f}(\boldsymbol{a_n}), \hat{f}(\boldsymbol{a_{n'}})]$ as shown in (3). The expectation $\boldsymbol{\Psi_f} = \mathbb{E}_{\mathrm{q}(\boldsymbol{a})}[\boldsymbol{K_{fu}}]$

connecting the alignments and the pseudo inputs is given by

$$\boldsymbol{\Psi_f} = \mathbb{E}_{\mathrm{q}(\boldsymbol{a})}[\boldsymbol{K_{fu}}], \text{ with}$$

$$(\boldsymbol{\Psi_f})_{ni} = \hat{\sigma}_{ni}^2 \sqrt{\frac{(\boldsymbol{\Sigma_a})_{nn}^{-1}}{\hat{\ell}_{ni} + (\boldsymbol{\Sigma_a})_{nn}^{-1}}} \exp\left(-\frac{1}{2}\frac{(\boldsymbol{\Sigma_a})_{nn}^{-1}\hat{\ell}_{ni}}{(\boldsymbol{\Sigma_a})_{nn}^{-1} + \hat{\ell}_{ni}}\left((\boldsymbol{\mu_a})_n - \boldsymbol{Z_i}\right)^2\right), \tag{7}$$

where $\hat{\ell}_{ni}$ is the combined length scale corresponding to the same kernel as $\hat{\sigma}_{ni}$. Lastly, $\boldsymbol{\Phi_f} = \mathbb{E}_{\mathrm{q}(\boldsymbol{a})}[\boldsymbol{K_{uf}K_{fu}}]$ connects alignments and pairs of pseudo inputs with the closed form

$$\boldsymbol{\Phi_f} = \mathbb{E}_{\mathrm{q}(\boldsymbol{a})}[\boldsymbol{K_{uf}K_{fu}}], \text{ with}$$

$$(\boldsymbol{\Phi_f})_{ij} = \sum_{n=1}^{N} \hat{\sigma}_{ni}^2\hat{\sigma}_{nj}^2 \sqrt{\frac{(\boldsymbol{\Sigma_a})_{nn}^{-1}}{\hat{\ell}_{ni} + \hat{\ell}_{nj} + (\boldsymbol{\Sigma_a})_{nn}^{-1}}} \exp\left(-\frac{1}{2}\frac{\hat{\ell}_{ni}\hat{\ell}_{nj}}{\hat{\ell}_{ni} + \hat{\ell}_{nj}}(\boldsymbol{Z_i} - \boldsymbol{Z_j})^2\right.$$

$$\left. -\frac{1}{2}\frac{(\boldsymbol{\Sigma_a})_{nn}^{-1}(\hat{\ell}_{ni} + \hat{\ell}_{nj})}{(\boldsymbol{\Sigma_a})_{nn}^{-1} + \hat{\ell}_{ni} + \hat{\ell}_{nj}}\left((\boldsymbol{\mu_a})_n - \frac{\hat{\ell}_{ni}\boldsymbol{Z_i} + \hat{\ell}_{nj}\boldsymbol{Z_j}}{\hat{\ell}_{ni} + \hat{\ell}_{nj}}\right)^2\right). \tag{8}$$

The Ψ-statistics factorize along the data and we only need to consider the diagonal entries of $\boldsymbol{\Sigma_a}$. If all the data belong to the same output, the Ψ-statistics of the squared exponential kernel can be recovered as a special case. This case is used for the output-specific warpings $\boldsymbol{g}$.

### 3.3 Model Interpretation

The graphical model shown in Figure 1 illustrates that the presented model can be interpreted as a group of $D$ deep GPs all of which share one layer which is a CP. This CP acts as an interface to share information between the different GPs which are otherwise conditionally independent. This modelling-choice introduces a new quality to the model when compared to standard deep GPs with multiple output dimensions, since the latter are not able in principle to learn dependencies between the different outputs. Compared to standard multi-output GPs, the AMO-GP introduces more flexibility with respect to the shared information. CPs make strong assumptions about the relative alignments of the different outputs, that is, they assume constant time-offsets. The AMO-GP extends this by introducing a principled Bayesian treatment of general nonlinear alignments $a_d$ on which we can place informative priors derived from the problem at hand. Together with the warping layers $g_d$, our model can learn to share knowledge in an informative latent space learnt from the data.

Alternatively, this model can be interpreted as a shared and warped latent variable model with a very specific prior: The indices $\boldsymbol{X}$ are part of the prior for the latent space $a_d(\boldsymbol{X})$ and specify a sense of order for the different data points $\boldsymbol{y}$ which is augmented with uncertainty by the alignment functions. Using this order, the convolution processes enforce the covariance structure for the different datapoints specified by the smoothing kernels.

In order to derive an inference scheme, we need the ability to propagate uncertainties about the correct alignments and latent shared information through subsequent layers. We adapted the approach of nested variational compression by Hensman and Lawrence [11], which is originally concerned with a single deep GP. The approximation is expanded to handle multiple GPs at once, yielding the bound in (5). The bound reflects the dependencies of the different outputs as the sharing of information between the different deep GPs is approximated through the shared inducing variables $\boldsymbol{u_{f,d}}$. Our main contribution for the inference scheme is the derivation of a closed-form solution for the Ψ-statistics of the convolution kernel in (6) to (8).

## 4 Experiments

In this section we show how to apply the AMO-GP to the task of finding common structure in time series observations. In this setting, we observe multiple time series $\mathcal{T}_d = (\boldsymbol{X_d}, \boldsymbol{y_d})$ and assume that there exist latent time series which determine the observations.

We will first apply the AMO-GP to an artificial data set in which we define a decomposed system of dependent time series by specifying a shared latent function generating the observations together

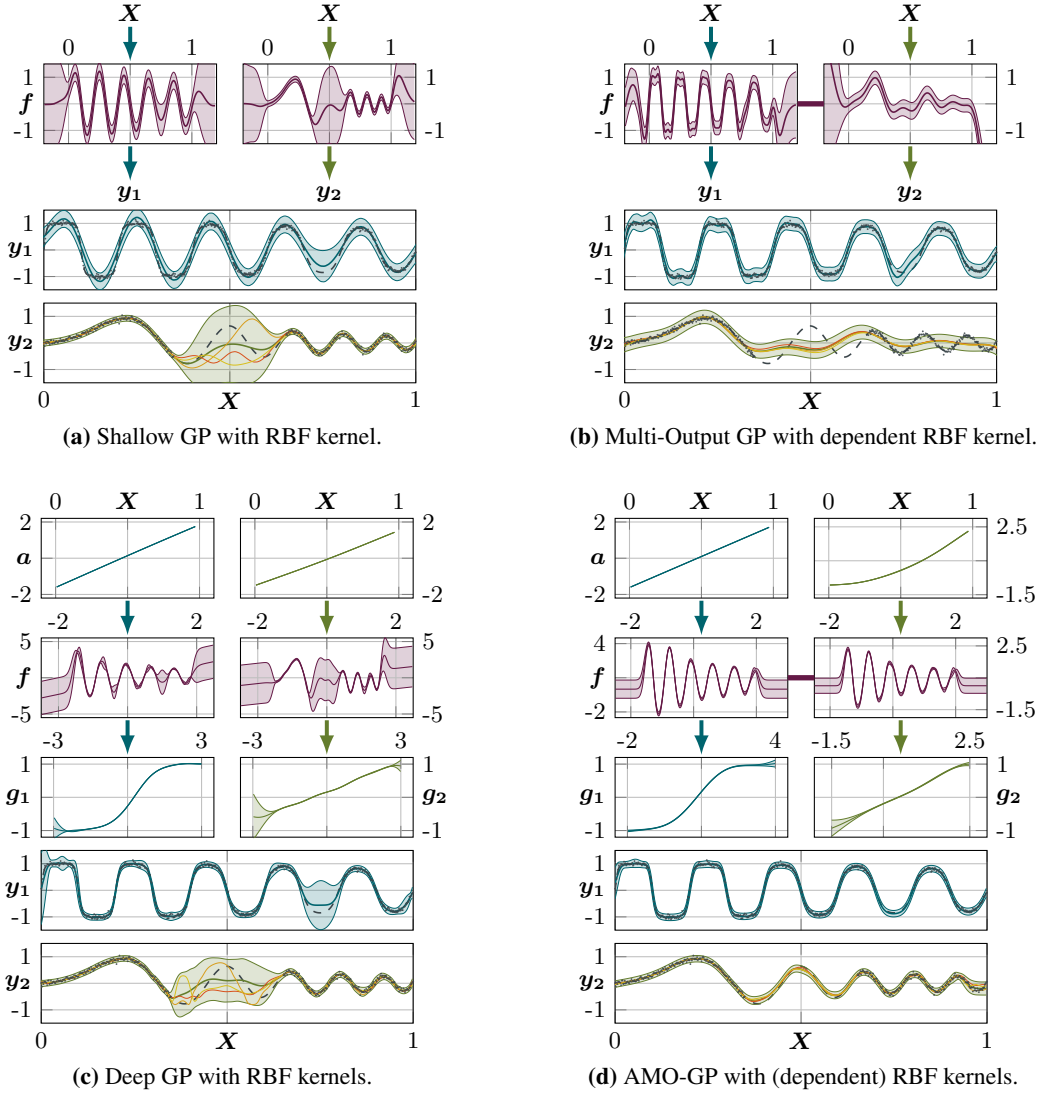

**Figure 3:** A comparison of the AMO-GP with other GP models. The plots show mean predictions and a shaded area of two standard deviations. If available, the ground truth is displayed as a dashed line. Additional lines are noiseless samples drawn from the model. The shallow and deep GPs in Figures 3a and 3c model the data independently and revert back to the prior in $y_2$. Because of the nonlinear alignment, a multi-output GP cannot model the data in Figure 3b. The AMO-GP in Figure 3d recovers the alignment and warping and shares information between the two outputs.

(a) Shallow GP with RBF kernel.

(b) Multi-Output GP with dependent RBF kernel.

(c) Deep GP with RBF kernels.

(d) AMO-GP with (dependent) RBF kernels.

with relative alignments and warpings for the different time series. We will show that our model is able to recover this decomposition from the training data and compare the results to other approaches of modeling the data. Then we focus on a real world data set of a neighbouring pair of wind turbines in a wind farm, where the model is able to recover a representation of the latent prevailing wind condition and the relative timings of wind fronts at the two turbines.

## 4.1 Artificial data set

Our data set consists of two time series $\mathcal{T}_1$ and $\mathcal{T}_2$ generated by a dampened sine function. We choose the alignment of $\mathcal{T}_1$ and the warping of $\mathcal{T}_2$ to be the identity in order to prevent us from directly observing the latent function and apply a sigmoid warping to $\mathcal{T}_1$. The alignment of $\mathcal{T}_2$ is selected to be a quadratic function. Figure 2 shows a visualization of this decomposed system of dependent time

**Table 1:** Test-log-likelihoods for the models presented in Section 4.

| Experiment | Test set | GP | MO-GP | DGP | AMO-GP (Ours) |
|---|---|---|---|---|---|
| Artificial | $[0.7, 0.8] \subseteq \mathcal{T}_1$ | -0.12 | -0.053 | 0.025 | **1.54** |
| | $[0.35, 0.65] \subseteq \mathcal{T}_2$ | -0.19 | -5.66 | -0.30 | **0.72** |
| Wind | $[40, 45] \subseteq \mathcal{T}_2$ | -4.42 | -2.31 | -1.80 | **-1.43** |
| | $[65, 75] \subseteq \mathcal{T}_2$ | -7.26 | -0.73 | -1.93 | **-0.69** |

series. To obtain training data we uniformly sampled 500 points from the two time series and added Gaussian noise. We subsequently removed parts of the training sets to explore the generalization behaviour of our model, resulting in $|\mathcal{T}_1| = 450$ and $|\mathcal{T}_2| = 350$.

We use this setup to train our model using squared exponential kernels both in the conditionally independent GPs $a_d$ and $g_d$ and as smoothing kernels in $f$. We can always choose one alignment and one warping to be the identity function in order to constrain the shared latent spaces $a$ and $f$ and provide a reference the other alignments and warpings will be relative to. Since we assume our artificial data simulates a physical system, we apply the prior knowledge that the alignment and warping processes have slower dynamics compared to the shared latent function which should capture most of the observed dynamics. To this end we applied priors to the $a_d$ and $g_d$ which prefer longer length scales and smaller variances compared to $f$. Otherwise, the model could easily get stuck in local minima like choosing the upper two layers to be identity functions and model the time series independently in the $g_d$. Additionally, our assumption of identity mean functions prevents pathological cases in which the complete model collapses to a constant function.

Figure 3d shows the AMO-GP's recovered function decomposition and joint predictions. The model successfully recovered a shared latent dampened sine function, a sigmoid warping for the first time series and an approximate quadratic alignment function for the second time series. In Figures 3a to 3c, we show the training results of a standard GP, a multi-output GP and a three-layer deep GP on the same data. For all of these models, we used RBF kernels and, in the case of the deep GP, applied priors similar to our model in order to avoid pathological cases. In Table 1 we report test log-likelihoods for the presented models, which illustrate the qualitative differences between the models. Because all models are non-parametric and converge well, repeating the experiments with different initializations leads to very similar likelihoods.

Both the standard GP and deep GP cannot learn dependencies between time series and revert back to the prior where no data is available. The deep GP has learned that two layers are enough to model the data and the resulting model is essentially a Bayesian warped GP which has identified the sigmoid warping for $\mathcal{T}_1$. Uncertainties in the deep GP are placed in the middle layer areas where no data are available for the respective time series, as sharing information between the two outputs is impossible. In contrast to the other two models, the multi-output GP can and must share information between the two time series. As discussed in Section 2 however, it is constrained to constant time-offsets and cannot model the nonlinear alignment in the data. Because of this, the model cannot recover the latent sine function and can only model one of the two outputs.

## 4.2 Pairs of wind turbines

This experiment is based on real data recorded from a pair of neighbouring wind turbines in a wind farm. The two time series $\mathcal{T}_1$ and $\mathcal{T}_2$ shown in gray in Figure 4 record the respective power generation of the two turbines over the course of one and a half hours, which was smoothed slightly using a rolling average over 60 seconds. There are 5400 data points for the first turbine (blue) and 4622 data points for the second turbine (green). We removed two intervals (drawn as dashed lines) from the second turbine's data set to inspect the behaviour of the model with missing data. This allows us to evaluate and compare the generative properties of our model in Figure 5.

The power generated by a wind turbine is mainly dependent on the speed of the wind fronts interacting with the turbine. For system identification tasks concerned with the behaviour of multiple wind turbines, associating the observations on different turbines due to the same wind fronts is an important task. However it is usually not possible to directly measure these correspondences or wind propagation

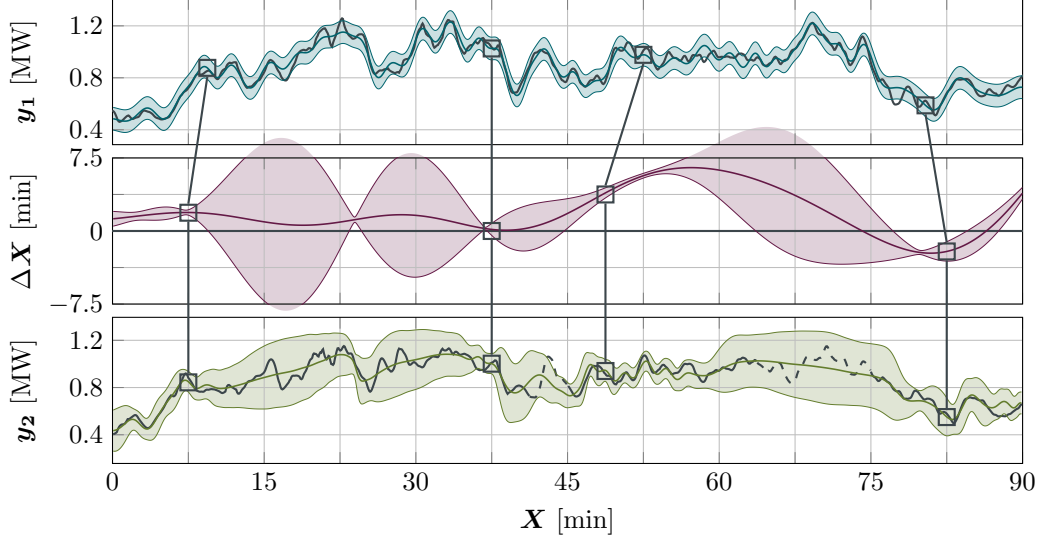

**Figure 4:** The joint posterior for two time series $y_1$ and $y_2$ of power production for a pair of wind turbines. The top and bottom plots show the two observed time series with training data and dashed missing data. The AMO-GP recovers an uncertain relative alignment of the two time series shown in the middle plot. High uncertainty about the alignment is placed in areas where multiple explanations are plausible due to the high amount of noise or missing data.

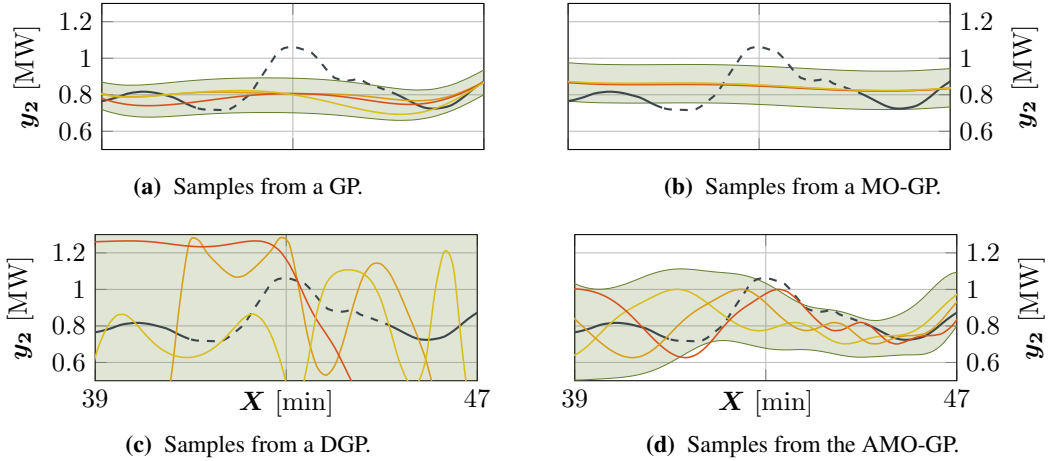

(a) Samples from a GP.

(b) Samples from a MO-GP.

(c) Samples from a DGP.

(d) Samples from the AMO-GP.

**Figure 5:** A comparison of noiseless samples drawn from a GP, a MO-GP, a DGP and the AMO-GP. The separation of uncertainties implied by the model structure of AMP-GP gives rise to an informative model. Since the uncertainty in the generative process is mainly placed in the relative alignment shown in Figure 4, all samples in Figure 5d resemble the underlying data in structure.

speeds between turbines, which means that there is no ground truth available. An additional problem is that the shared latent wind conditions are superimposed by turbine-specific local turbulences. Since these local effects are of comparable amplitude to short-term changes of wind speed, it is challenging to decide which parts of the signal to explain away as noise and which part to identify as the underlying shared process.

Our goal is the simultaneous learning of the uncertain alignment in time $a$ and of the shared latent wind condition $f$. Modelling the turbine-specific parts of the signals is not the objective, so they need to be explained by the Gaussian noise term. We use a squared exponential kernel as a prior for the alignment functions $a_d$ and as smoothening kernels in $f$. For the given data set we can assume the output warpings $g_d$ to be linear functions because there is only one dimension, the power generation, which in this data set is of similar shape for both turbines. Again we encode a preference

for alignments with slow dynamics with a prior on the length scales of $a_d$. As the signal has turbine-specific autoregressive components, plausible alignments are not unique. To constrain the AMO-GP, we want it to prefer alignments close to the identity function which we chose as a prior mean function.

Figure 4 shows the joint model learned from the data in which $a_1$ is chosen to be the identity function. The possible alignments identified match the physical conditions of the wind farm. For the given turbines, time offsets of up to six minutes are plausible and for most wind conditions, the offset is expected to be close to zero. For areas where the alignment is quite certain however, the two time series are explained with comparable detail. The model is able to recover unambiguous associations well and successfully places high uncertainty on the alignment in areas where multiple explanations are plausible due to the noisy signal.

As expected, the uncertainty about the alignment also grows where data for the second time series is missing. This uncertainty is propagated through the shared function and results in higher predictive variances for the second time series. Because of the factorization in the model however, we can recover the uncertainties about the alignment and the shared latent function separately. Figure 5 compares samples drawn from our model with samples drawn from a GP, a MO-GP and a DGP. The GP reverts to their respective priors when data is missing, while the MO-GP does not handle short-term dynamics and smoothens the signal enough such that the nonlinear alignment can be approximated as constant. Samples drawn from a DGP model showcase the complexity of a DGP prior. Unconstrained composite GPs are hard to reason about and make the model very flexible in terms of representable functions. Since the model's evidence is very broad, the posterior is uninformed and inference is hard. Additionally, as discussed in Appendix B and [11], the nested variational compression bound tends to loosen with high uncertainties. AMO-GP shows richer structure: Due to the constraints imposed by the model, more robust inference leads to a more informed model. Samples show that it has learned that a maximum which is missing in the training data has to exist somewhere, but the uncertainty about the correct alignment due to the local turbulence means that different samples place the maximum at different locations in $\boldsymbol{X}$-direction.

## 5    Conclusion

We have proposed the warped and aligned multi-output Gaussian process (AMO-GP), in which MO-GPs are embedded in a hierarchy to find shared structure in latent spaces. We extended convolution processes [5] with conditionally independent Gaussian processes on both the input and output sides, giving rise to a highly structured deep GP model. This structure can be used to both regularize the model and encode expert knowledge about specific parts of the system. By applying nested variational compression [11] to inference in these models, we presented a variational lower bound which combines Bayesian treatment of all parts of the model with scalability via stochastic optimization.

We compared the model with GPs, deep GPs and multi-output GPs on an artificial data set and showed how the richer model-structure allows the AMO-GP to pick up on latent structure which the other approaches cannot model. We then applied the AMO-GP to real world data of two wind turbines and used the proposed hierarchy to model wind propagation in a wind farm and recover information about the latent non homogeneous wind field. With uncertainties decomposed along the hierarchy, our approach handles ambiguities introduced by the stochasticity of the wind in a principled manner. This indicates the AMO-GP is a good approach for these kinds of dynamical system, where multiple misaligned sensors measure the same latent effect.

## 6    Acknowledgement

The project this report is based on was supported with funds from the German Federal Ministry of Education and Research under project number 01IB15001. The sole responsibility for the reports contents lies with the authors.

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
