[Supplementary Material]

# A Detailed Variational Approximation

In this section, we repeat the derivation of the variational approximation in more detail.

Since exact inference in this model is intractable, we discuss a variational approximation to the model's true marginal likelihood and posterior in this section. Analogously to $\boldsymbol{y}$, we denote the random vectors which contain the function values of the respective functions and outputs as $\boldsymbol{a}$ and $\boldsymbol{f}$. The joint probability distribution of the data can then be written as

$$\mathrm{p}(\boldsymbol{y}, \boldsymbol{f}, \boldsymbol{a} \,|\, \boldsymbol{X}) = \mathrm{p}(\boldsymbol{f} \,|\, \boldsymbol{a}) \prod_{d=1}^{D} \mathrm{p}(\boldsymbol{y_d} \,|\, \boldsymbol{f_d}) \, \mathrm{p}(\boldsymbol{a_d} \,|\, \boldsymbol{X}),$$
$$\boldsymbol{a_d} \,|\, \boldsymbol{X} \sim \mathcal{N}(\boldsymbol{X}, \boldsymbol{K_{a,d}} + \sigma_{a,d}^2 \mathbf{I}), \qquad (9)$$
$$\boldsymbol{f} \,|\, \boldsymbol{a} \sim \mathcal{N}(\boldsymbol{0}, \boldsymbol{K_f} + \sigma_f^2 \mathbf{I}),$$
$$\boldsymbol{y_d} \,|\, \boldsymbol{f_d} \sim \mathcal{N}(\boldsymbol{f_d}, \boldsymbol{K_{g,d}} + \sigma_{y,d}^2 \mathbf{I}).$$

Here, we use $\boldsymbol{K}$ to refer to the Gram matrix corresponding to the kernel of the respective GP. All but the CPs factorize over both the different levels of the model as well as the different outputs.

To approximate a single deep GP, Hensman and Lawrence [11] proposed nested variational compression in which every GP in the hierarchy is handled independently. While this forces a variational approximation of all intermediate outputs of the stacked processes, it has the appealing property that it allows optimization via stochastic gradient descent [10] and the variational approximation can after training be used independently of the original training data.

## A.1  Augmented Model

Nested variational compression focuses on augmenting a full GP model by introducing sets of *inducing variables* $\boldsymbol{u}$ with their *inducing inputs* $\boldsymbol{Z}$. Those variables are assumed to be latent observations of the same functions and are thus jointly Gaussian with the observed data.

It can be written using its marginals [22] as

$$\mathrm{p}(\hat{\boldsymbol{a}}, \boldsymbol{u}) = \mathcal{N}(\hat{\boldsymbol{a}} \,|\, \boldsymbol{\mu_a}, \boldsymbol{\Sigma_a}) \, \mathcal{N}(\boldsymbol{u} \,|\, \boldsymbol{Z}, \boldsymbol{K_{uu}}), \text{ with}$$
$$\boldsymbol{\mu_a} = \boldsymbol{X} + \boldsymbol{K_{au}} \boldsymbol{K_{uu}^{-1}} (\boldsymbol{u} - \boldsymbol{Z}), \qquad (10)$$
$$\boldsymbol{\Sigma_a} = \boldsymbol{K_{aa}} - \boldsymbol{K_{au}} \boldsymbol{K_{uu}^{-1}} \boldsymbol{K_{ua}},$$

where, after dropping some indices and explicit conditioning on $\boldsymbol{X}$ and $\boldsymbol{Z}$ for clarity, $\hat{\boldsymbol{a}}$ denotes the function values $a_d(\boldsymbol{X})$ without noise and we write the Gram matrices as $\boldsymbol{K_{au}} = k_{a,d}(\boldsymbol{X}, \boldsymbol{Z})$.

While the original model in (9) can be recovered exactly by marginalizing the inducing variables, considering a specific variational approximation of the joint $\mathrm{p}(\hat{\boldsymbol{a}}, \boldsymbol{u})$ gives rise to the desired lower bound in the next subsection. A central assumption of this approximation [22] is that given enough inducing variables at the correct location, they are a sufficient statistic for $\hat{\boldsymbol{a}}$, implying conditional independence of the entries of $\hat{\boldsymbol{a}}$ given $\boldsymbol{X}$ and $\boldsymbol{u}$. We introduce such inducing variables for every GP in the model, yielding the set $\{\boldsymbol{u_{a,d}}, \boldsymbol{u_{f,d}}, \boldsymbol{u_{g,d}}\}_{d=1}^{D}$ of inducing variables. Note that for the CP $f$, we introduce one set of inducing variables $\boldsymbol{u_{f,d}}$ per output $f_d$. These inducing variables play a crucial role in sharing information between the different outputs.

## A.2  Variational Lower Bound

To derive the desired variational lower bound for the log marginal likelihood of the complete model, multiple steps are necessary. First, we will consider the innermost GPs $a_d$ describing the alignment functions. We derive the Scalable Variational GP (SVGP), a lower bound for this model part which can be calculated efficiently and can be used for stochastic optimization, first introduced by Hensman, Fusi, and Lawrence [10]. In order to apply this bound recursively, we will both show how to propagate the uncertainty through the subsequent layers $f_d$ and $g_d$ and how to avoid the inter-layer cross-dependencies using another variational approximation as presented by Hensman and Lawrence [11]. While Hensman and Lawrence considered standard deep GP models, we will show how to apply their results to CPs.

**The First Layer**  Since the inputs $\boldsymbol{X}$ are fully known, we do not need to propagate uncertainty through the GPs $a_d$. Instead, the uncertainty about the $\boldsymbol{a_d}$ comes from the uncertainty about the correct functions $a_d$ and is introduced by the processes themselves. To derive a lower bound on the marginal log likelihood of $\boldsymbol{a_d}$, we assume a variational distribution $\mathrm{q}(\boldsymbol{u_{a,d}}) \sim \mathcal{N}(\boldsymbol{m_{a,d}}, \boldsymbol{S_{a,d}})$ approximating $\mathrm{p}(\boldsymbol{u_{a,d}})$ and additionally assume that $\mathrm{q}(\hat{\boldsymbol{a}}_d, \boldsymbol{u_{a,d}}) = \mathrm{p}(\hat{\boldsymbol{a}}_d \,|\, \boldsymbol{u_{a,d}})\, \mathrm{q}(\boldsymbol{u_{a,d}})$. After dropping the indices again, using Jensen's inequality we get

$$
\begin{aligned}
\log \mathrm{p}(\boldsymbol{a} \,|\, \boldsymbol{X}) &= \log \int \mathrm{p}(\boldsymbol{a} \,|\, \boldsymbol{u})\, \mathrm{p}(\boldsymbol{u})\, \mathrm{d}\boldsymbol{u} \\
&= \log \int \mathrm{q}(\boldsymbol{u}) \frac{\mathrm{p}(\boldsymbol{a} \,|\, \boldsymbol{u})\, \mathrm{p}(\boldsymbol{u})}{\mathrm{q}(\boldsymbol{u})}\, \mathrm{d}\boldsymbol{u} \\
&\geq \int \mathrm{q}(\boldsymbol{u}) \log \frac{\mathrm{p}(\boldsymbol{a} \,|\, \boldsymbol{u})\, \mathrm{p}(\boldsymbol{u})}{\mathrm{q}(\boldsymbol{u})}\, \mathrm{d}\boldsymbol{u} \\
&= \int \log \mathrm{p}(\boldsymbol{a} \,|\, \boldsymbol{u})\, \mathrm{q}(\boldsymbol{u})\, \mathrm{d}\boldsymbol{u} - \int \mathrm{q}(\boldsymbol{u}) \log \frac{\mathrm{q}(\boldsymbol{u})}{\mathrm{p}(\boldsymbol{u})}\, \mathrm{d}\boldsymbol{u} \\
&= \mathbb{E}_{\mathrm{q}(\boldsymbol{u})}[\log \mathrm{p}(\boldsymbol{a} \,|\, \boldsymbol{u})] - \mathrm{KL}(\mathrm{q}(\boldsymbol{u}) \,\|\, \mathrm{p}(\boldsymbol{u})),
\end{aligned}
\tag{11}
$$

where $\mathbb{E}_{\mathrm{q}(\boldsymbol{u})}[\,\cdot\,]$ denotes the expected value with respect to the distribution $\mathrm{q}(\boldsymbol{u})$ and $\mathrm{KL}(\,\cdot\,\|\,\cdot\,)$ denotes the KL divergence, which can be evaluated analytically.

To bound the required expectation, we use Jensen's inequality again together with (10) which gives

$$
\begin{aligned}
\log \mathrm{p}(\boldsymbol{a} \,|\, \boldsymbol{u}) &= \log \int \mathrm{p}(\boldsymbol{a} \,|\, \hat{\boldsymbol{a}})\, \mathrm{p}(\hat{\boldsymbol{a}} \,|\, \boldsymbol{u})\, \mathrm{d}\hat{\boldsymbol{a}} \\
&= \log \int \mathcal{N}(\boldsymbol{a} \,|\, \hat{\boldsymbol{a}}, \sigma_a^2 \mathbf{I})\, \mathcal{N}(\hat{\boldsymbol{a}} \,|\, \boldsymbol{\mu_a}, \boldsymbol{\Sigma_a})\, \mathrm{d}\hat{\boldsymbol{a}} \\
&\geq \int \log \mathcal{N}(\boldsymbol{a} \,|\, \hat{\boldsymbol{a}}, \sigma_a^2 \mathbf{I})\, \mathcal{N}(\hat{\boldsymbol{a}} \,|\, \boldsymbol{\mu_a}, \boldsymbol{\Sigma_a})\, \mathrm{d}\hat{\boldsymbol{a}} \\
&= \log \mathcal{N}(\boldsymbol{a} \,|\, \boldsymbol{\mu_a}, \sigma_a^2 \mathbf{I}) - \frac{1}{2\sigma_a^2} \mathrm{tr}(\boldsymbol{\Sigma_a}).
\end{aligned}
\tag{12}
$$

We apply this bound to the expectation to get

$$
\mathbb{E}_{\mathrm{q}(\boldsymbol{u})}[\log \mathrm{p}(\boldsymbol{a} \,|\, \boldsymbol{u})] \geq \mathbb{E}_{\mathrm{q}(\boldsymbol{u})}[\log \mathcal{N}(\boldsymbol{a} \,|\, \boldsymbol{\mu_a}, \sigma_a^2 \mathbf{I})] - \frac{1}{2\sigma_a^2} \mathrm{tr}(\boldsymbol{\Sigma_a}), \text{ with} \tag{13}
$$

$$
\begin{aligned}
\mathbb{E}_{\mathrm{q}(\boldsymbol{u})}[\log \mathcal{N}(\boldsymbol{a} \,|\, \boldsymbol{\mu_a}, \sigma_a^2 \mathbf{I})] &= \log \mathcal{N}(\boldsymbol{a} \,|\, \boldsymbol{K_{au}} \boldsymbol{K_{uu}^{-1}} \boldsymbol{m}, \sigma_a^2 \mathbf{I}) \\
&\quad + \frac{1}{2\sigma_a^2} \mathrm{tr}\big( \boldsymbol{K_{au}} \boldsymbol{K_{uu}^{-1}} \boldsymbol{S} \boldsymbol{K_{uu}^{-1}} \boldsymbol{K_{ua}} \big).
\end{aligned}
\tag{14}
$$

Resubstituting this result into (11) yields the final bound

$$
\begin{aligned}
\log \mathrm{p}(\boldsymbol{a} \,|\, \boldsymbol{X}) \geq{}& \log \mathcal{N}(\boldsymbol{a} \,|\, \boldsymbol{K_{au}} \boldsymbol{K_{uu}^{-1}} \boldsymbol{m}, \sigma_a^2 \mathbf{I}) - \mathrm{KL}(\mathrm{q}(\boldsymbol{u}) \,\|\, \mathrm{p}(\boldsymbol{u})) \\
&- \frac{1}{2\sigma_a^2} \mathrm{tr}(\boldsymbol{\Sigma_a}) - \frac{1}{2\sigma_a^2} \mathrm{tr}\big( \boldsymbol{K_{au}} \boldsymbol{K_{uu}^{-1}} \boldsymbol{S} \boldsymbol{K_{uu}^{-1}} \boldsymbol{K_{ua}} \big).
\end{aligned}
\tag{15}
$$

This bound, which depends on the hyper parameters of the kernel and likelihood $\{\boldsymbol{\theta}, \sigma_a\}$ and the variational parameters $\{\boldsymbol{Z}, \boldsymbol{m}, \boldsymbol{S}\}$, can be calculated in $\mathcal{O}(NM^2)$ time. It factorizes along the data points which enables stochastic optimization.

In order to obtain a bound on the full model, we apply the same techniques to the other processes. Since the alignment processes $a_d$ are assumed to be independent, we have $\log \mathrm{p}(\boldsymbol{a_1}, \ldots, \boldsymbol{a_D} \,|\, \boldsymbol{X}) = \sum_{d=1}^{D} \log \mathrm{p}(\boldsymbol{a_d} \,|\, \boldsymbol{X})$, where every term can be approximated using the bound in (15). However, for all subsequent layers, the bound is not directly applicable, since the inputs are no longer known but instead are given by the outputs of the previous process. It is therefore necessary to propagate their uncertainty and also handle the interdependencies between the layers introduced by the latent function values $\boldsymbol{a}$, $\boldsymbol{f}$ and $\boldsymbol{g}$.

**The Second and Third Layer**    Our next goal is to derive a bound on the outputs of the second layer

$$\log \mathrm{p}(\boldsymbol{f} \mid \boldsymbol{u_f}) = \log \int \mathrm{p}(\boldsymbol{f}, \boldsymbol{a}, \boldsymbol{u_a} \mid \boldsymbol{u_f}) \, \mathrm{d}\boldsymbol{a} \, \mathrm{d}\boldsymbol{u_a}, \tag{16}$$

that is, an expression in which the uncertainty about the different $\boldsymbol{a_d}$ and the cross-layer dependencies on the $\boldsymbol{u_{a,d}}$ are both marginalized. While on the first layer, the different $\boldsymbol{a_d}$ are conditionally independent, the second layer explicitly models the cross-covariances between the different outputs via convolutions over the shared latent processes $w_r$. We will therefore need to handle all of the different $\boldsymbol{f_d}$, together denoted as $\boldsymbol{f}$, at the same time.

We start by considering the relevant terms from (9) and apply (12) to marginalize $\boldsymbol{a}$ in

$$
\begin{aligned}
\log \mathrm{p}(\boldsymbol{f} \mid \boldsymbol{u_f}, \boldsymbol{u_a}) &= \log \int \mathrm{p}(\boldsymbol{f}, \boldsymbol{a} \mid \boldsymbol{u_f}, \boldsymbol{u_a}) \, \mathrm{d}\boldsymbol{a} \\
&\geq \log \int \tilde{\mathrm{p}}(\boldsymbol{f} \mid \boldsymbol{u_f}, \boldsymbol{a}) \tilde{\mathrm{p}}(\boldsymbol{a} \mid \boldsymbol{u_a}) \cdot \exp\left( -\frac{1}{2\sigma_a^2} \operatorname{tr}(\boldsymbol{\Sigma_a}) - \frac{1}{2\sigma_f^2} \operatorname{tr}(\boldsymbol{\Sigma_f}) \right) \mathrm{d}\boldsymbol{a} \\
&\geq \mathbb{E}_{\tilde{\mathrm{p}}(\boldsymbol{a}|\boldsymbol{u_a})}[\log \tilde{\mathrm{p}}(\boldsymbol{f} \mid \boldsymbol{u_f}, \boldsymbol{a})] - \mathbb{E}_{\tilde{\mathrm{p}}(\boldsymbol{a}|\boldsymbol{u_a})}\left[ \frac{1}{2\sigma_f^2} \operatorname{tr}(\boldsymbol{\Sigma_f}) \right] - \frac{1}{2\sigma_a^2} \operatorname{tr}(\boldsymbol{\Sigma_a}),
\end{aligned} \tag{17}
$$

where we write $\tilde{\mathrm{p}}(\boldsymbol{a} \mid \boldsymbol{u_a}) = \mathcal{N}\left(\boldsymbol{a} \mid \boldsymbol{\mu_a}, \sigma_a^2 \mathbf{I}\right)$ to incorporate the Gaussian noise in the latent space. Due to our assumption that $\boldsymbol{u_a}$ is a sufficient statistic for $\boldsymbol{a}$ we choose

$$
\begin{aligned}
\mathrm{q}(\boldsymbol{a} \mid \boldsymbol{u_a}) &= \tilde{\mathrm{p}}(\boldsymbol{a} \mid \boldsymbol{u_a}), \text{ and} \\
\mathrm{q}(\boldsymbol{a}) &= \int \tilde{\mathrm{p}}(\boldsymbol{a} \mid \boldsymbol{u_a}) \, \mathrm{q}(\boldsymbol{u_a}) \, \mathrm{d}\boldsymbol{u_a},
\end{aligned} \tag{18}
$$

and use another variational approximation to marginalize $\boldsymbol{u_a}$. This yields

$$
\begin{aligned}
\log \mathrm{p}(\boldsymbol{f} \mid \boldsymbol{u_f}) &= \log \int \mathrm{p}(\boldsymbol{f}, \boldsymbol{u_a} \mid \boldsymbol{u_f}) \, \mathrm{d}\boldsymbol{u_a} \\
&= \log \int \mathrm{p}(\boldsymbol{f} \mid \boldsymbol{u_f}, \boldsymbol{u_a}) \, \mathrm{p}(\boldsymbol{u_a}) \, \mathrm{d}\boldsymbol{u_a} \\
&\geq \int \mathrm{q}(\boldsymbol{u_a}) \log \frac{\mathrm{p}(\boldsymbol{f} \mid \boldsymbol{u_f}, \boldsymbol{u_a}) \, \mathrm{p}(\boldsymbol{u_a})}{\mathrm{q}(\boldsymbol{u_a})} \, \mathrm{d}\boldsymbol{u_a} \\
&= \mathbb{E}_{\mathrm{q}(\boldsymbol{u_a})}[\log \mathrm{p}(\boldsymbol{f} \mid \boldsymbol{u_a}, \boldsymbol{u_f})] - \mathrm{KL}(\mathrm{q}(\boldsymbol{u_a}) \,\|\, \mathrm{p}(\boldsymbol{u_a})) \\
&\geq \mathbb{E}_{\mathrm{q}(\boldsymbol{u_a})}\big[\mathbb{E}_{\tilde{\mathrm{p}}(\boldsymbol{a}|\boldsymbol{u_a})}[\log \tilde{\mathrm{p}}(\boldsymbol{f} \mid \boldsymbol{u_f}, \boldsymbol{a})]\big] - \mathrm{KL}(\mathrm{q}(\boldsymbol{u_a}) \,\|\, \mathrm{p}(\boldsymbol{u_a})) \\
&\quad - \frac{1}{2\sigma_a^2} \operatorname{tr}(\boldsymbol{\Sigma_a}) - \mathbb{E}_{\mathrm{q}(\boldsymbol{u_a})}\left[ \mathbb{E}_{\tilde{\mathrm{p}}(\boldsymbol{a}|\boldsymbol{u_a})}\left[ \frac{1}{2\sigma_f^2} \operatorname{tr}(\boldsymbol{\Sigma_f}) \right] \right] \\
&\geq \mathbb{E}_{\mathrm{q}(\boldsymbol{a})}[\log \tilde{\mathrm{p}}(\boldsymbol{f} \mid \boldsymbol{u_f}, \boldsymbol{a})], - \mathrm{KL}(\mathrm{q}(\boldsymbol{u_a}) \,\|\, \mathrm{p}(\boldsymbol{u_a})) \\
&\quad - \frac{1}{2\sigma_a^2} \operatorname{tr}(\boldsymbol{\Sigma_a}) - \frac{1}{2\sigma_f^2} \mathbb{E}_{\mathrm{q}(\boldsymbol{a})}[\operatorname{tr}(\boldsymbol{\Sigma_f})],
\end{aligned} \tag{19}
$$

where we apply Fubini's theorem to exchange the order of integration in the expected values. The expectations with respect to $\mathrm{q}(\boldsymbol{a})$ involve expectations of kernel matrices, also called $\Psi$-statistics, in the same way as in [8] and are given by

$$
\begin{aligned}
\psi_f &= \mathbb{E}_{\mathrm{q}(\boldsymbol{a})}[\operatorname{tr}(\boldsymbol{K_{ff}})], \\
\boldsymbol{\Psi_f} &= \mathbb{E}_{\mathrm{q}(\boldsymbol{a})}[\boldsymbol{K_{fu}}], \\
\boldsymbol{\Phi_f} &= \mathbb{E}_{\mathrm{q}(\boldsymbol{a})}[\boldsymbol{K_{uf}} \boldsymbol{K_{fu}}].
\end{aligned} \tag{20}
$$

These $\Psi$-statistics can be computed analytically for multiple kernels, including the squared exponential kernel. In Appendix A.3 we show closed-form solutions for these $\Psi$-statistics for the implicit kernel defined in the CP layer. To obtain the final formulation of the desired bound for $\log \mathrm{p}(\boldsymbol{f} \mid \boldsymbol{u_f})$

we substitute (20) into (19) and get the analytically tractable bound

$$\log \mathrm{p}(\boldsymbol{f} \mid \boldsymbol{u_f}) \geq \log \mathcal{N}\left(\boldsymbol{f} \;\middle|\; \boldsymbol{\Psi_f} \boldsymbol{K}_{\boldsymbol{u_f u_f}}^{-1} \boldsymbol{m_f}, \sigma_f^2 \mathbf{I}\right) - \mathrm{KL}(\mathrm{q}(\boldsymbol{u_a}) \,\|\, \mathrm{p}(\boldsymbol{u_a})) - \frac{1}{2\sigma_a^2}\,\mathrm{tr}(\boldsymbol{\Sigma_a})$$
$$- \frac{1}{2\sigma_f^2}\left(\psi_f - \mathrm{tr}\left(\boldsymbol{\Psi_f} \boldsymbol{K}_{\boldsymbol{u_f u_f}}^{-1}\right)\right) \tag{21}$$
$$- \frac{1}{2\sigma_f^2}\,\mathrm{tr}\left(\left(\boldsymbol{\Phi_f} - \boldsymbol{\Psi_f}^\top \boldsymbol{\Psi_f}\right) \boldsymbol{K}_{\boldsymbol{u_f u_f}}^{-1}\left(\boldsymbol{m_f m_f}^\top + \boldsymbol{S_f}\right) \boldsymbol{K}_{\boldsymbol{u_f u_f}}^{-1}\right)$$

The uncertainties in the first layer have been propagated variationally to the second layer. Besides the regularization terms, $\boldsymbol{f} \mid \boldsymbol{u_f}$ is a Gaussian distribution. Because of their cross dependencies, the different outputs $\boldsymbol{f_d}$ are considered in a common bound and do not factorize along dimensions. The third layer warpings $\boldsymbol{g_d}$ however are conditionally independent given $\boldsymbol{f}$ and can therefore be considered separately. In order to derive a bound for $\log \mathrm{p}(\boldsymbol{y} \mid \boldsymbol{u_g})$ we apply the same steps as described above, resulting in the final bound, which factorizes along the data, allowing for stochastic optimization methods:

$$\log \mathrm{p}(\boldsymbol{y} \mid \boldsymbol{X}) \geq \sum_{d=1}^{D} \log \mathcal{N}\left(\boldsymbol{y_d} \;\middle|\; \boldsymbol{\Psi_{g,d}} \boldsymbol{K}_{\boldsymbol{u_{g,d} u_{g,d}}}^{-1} \boldsymbol{m_{g,d}}, \sigma_{y,d}^2 \mathbf{I}\right) - \sum_{d=1}^{D} \frac{1}{2\sigma_{a,d}^2}\,\mathrm{tr}(\boldsymbol{\Sigma_{a,d}})$$

$$- \frac{1}{2\sigma_f^2}\left(\psi_f - \mathrm{tr}\left(\boldsymbol{\Phi_f} \boldsymbol{K}_{\boldsymbol{u_f u_f}}^{-1}\right)\right) - \sum_{d=1}^{D} \frac{1}{2\sigma_{y,d}^2}\left(\psi_{g,d} - \mathrm{tr}\left(\boldsymbol{\Phi_{g,d}} \boldsymbol{K}_{\boldsymbol{u_{g,d} u_{g,d}}}^{-1}\right)\right)$$

$$- \sum_{d=1}^{D} \mathrm{KL}(\mathrm{q}(\boldsymbol{u_{a,d}}) \,\|\, \mathrm{p}(\boldsymbol{u_{a,d}})) - \mathrm{KL}(\mathrm{q}(\boldsymbol{u_f}) \,\|\, \mathrm{p}(\boldsymbol{u_f})) - \sum_{d=1}^{D} \mathrm{KL}(\mathrm{q}(\boldsymbol{u_{y,d}}) \,\|\, \mathrm{p}(\boldsymbol{u_{y,d}})) \tag{22}$$

$$- \frac{1}{2\sigma_f^2}\,\mathrm{tr}\left(\left(\boldsymbol{\Phi_f} - \boldsymbol{\Psi_f}^\top \boldsymbol{\Psi_f}\right) \boldsymbol{K}_{\boldsymbol{u_f u_f}}^{-1}\left(\boldsymbol{m_f m_f}^\top + \boldsymbol{S_f}\right) \boldsymbol{K}_{\boldsymbol{u_f u_f}}^{-1}\right)$$

$$- \sum_{d=1}^{D} \frac{1}{2\sigma_{y,d}^2}\,\mathrm{tr}\left(\left(\boldsymbol{\Phi_{g,d}} - \boldsymbol{\Psi_{g,d}}^\top \boldsymbol{\Psi_{g,d}}\right) \boldsymbol{K}_{\boldsymbol{u_{g,d} u_{g,d}}}^{-1}\left(\boldsymbol{m_{g,d} m_{g,d}}^\top + \boldsymbol{S_{g,d}}\right) \boldsymbol{K}_{\boldsymbol{u_{g,d} u_{g,d}}}^{-1}\right)$$

### A.3 Convolution Kernel Expectations

In Section 2 we assumed the latent processes $w_r$ to be white noise processes and the smoothing kernel functions $T_{d,r}$ to be squared exponential kernels, leading to an explicit closed form formulation for the covariance between outputs shown in (3). In this section, we derive the $\Psi$-statistics for this generalized squared exponential kernel needed to evaluate (22).

The uncertainty about the first layer is captured by the variational distribution of the latent alignments $\boldsymbol{a}$ given by $\mathrm{q}(\boldsymbol{a}) \sim \mathcal{N}(\boldsymbol{\mu_a}, \boldsymbol{\Sigma_a})$, with $\boldsymbol{a} = (\boldsymbol{a_1}, \ldots, \boldsymbol{a_d})$. Every aligned point in $\boldsymbol{a}$ corresponds to one output of $\boldsymbol{f}$ and ultimately to one of the $\boldsymbol{y_d}$. Since the closed form of the multi output kernel depends on the choice of outputs, we will use the notation $\hat{f}(\boldsymbol{a_n})$ to denote $f_d(\boldsymbol{a_n})$ such that $\boldsymbol{a_n}$ is associated with output $d$.

For notational simplicity, we only consider the case of one single latent process $w_r$. Since the latent processes are independent, the results can easily be generalized to multiple processes. Then, $\psi_f$ is given by

$$\psi_f = \mathbb{E}_{\mathrm{q}(\boldsymbol{a})}[\mathrm{tr}(\boldsymbol{K_{ff}})]$$
$$= \sum_{n=1}^{N} \mathbb{E}_{\mathrm{q}(\boldsymbol{a_n})}\left[\mathrm{cov}\left[\hat{f}(\boldsymbol{a_n}), \hat{f}(\boldsymbol{a_n})\right]\right]$$
$$= \sum_{n=1}^{N} \int \mathrm{cov}\left[\hat{f}(\boldsymbol{a_n}), \hat{f}(\boldsymbol{a_n})\right] \mathrm{q}(\boldsymbol{a_n})\,\mathrm{d}\boldsymbol{a_n} \tag{23}$$
$$= \sum_{n=1}^{N} \hat{\sigma}_{nn}^2.$$

Similar to the notation $\hat{f}(\cdot)$, we use the notation $\hat{\sigma}_{nn'}$ to mean the variance term associated with the covariance function $\text{cov}[\hat{f}(\boldsymbol{a_n}), \hat{f}(\boldsymbol{a_{n'}})]$. The expectation $\boldsymbol{\Psi_f} = \mathbb{E}_{\text{q}(\boldsymbol{a})}[\boldsymbol{K_{fu}}]$ connecting the alignments and the pseudo inputs is given by

$$\boldsymbol{\Psi_f} = \mathbb{E}_{\text{q}(\boldsymbol{a})}[\boldsymbol{K_{fu}}], \text{ with}$$

$$(\boldsymbol{\Psi_f})_{ni} = \int \text{cov}\Big[\hat{f}(\boldsymbol{a_n}), \hat{f}(\boldsymbol{Z_i})\Big]\, \text{q}(\boldsymbol{a_n})\, \text{d}\boldsymbol{a_n}$$

$$= \hat{\sigma}_{ni}^2 \sqrt{\frac{(\boldsymbol{\Sigma_a})_{nn}^{-1}}{\hat{\ell}_{ni} + (\boldsymbol{\Sigma_a})_{nn}^{-1}}} \cdot \exp\left(-\frac{1}{2}\frac{(\boldsymbol{\Sigma_a})_{nn}^{-1}\hat{\ell}_{ni}}{(\boldsymbol{\Sigma_a})_{nn}^{-1} + \hat{\ell}_{ni}}\left((\boldsymbol{\mu_a})_n - \boldsymbol{Z_i}\right)^2\right)$$

(24)

where $\hat{\ell}_{ni}$ is the combined length scale corresponding to the same kernel as $\hat{\sigma}_{ni}$. Lastly, $\boldsymbol{\Phi_f} = \mathbb{E}_{\text{q}(\boldsymbol{a})}[\boldsymbol{K_{uf}}\boldsymbol{K_{fu}}]$ connects alignments and pairs of pseudo inputs with the closed form

$$\boldsymbol{\Phi_f} = \mathbb{E}_{\text{q}(\boldsymbol{a})}[\boldsymbol{K_{uf}}\boldsymbol{K_{fu}}], \text{ with}$$

$$(\boldsymbol{\Phi_f})_{ij} = \sum_{n=1}^{N} \int \text{cov}\Big[\hat{f}(\boldsymbol{a_n}), \hat{f}(\boldsymbol{Z_i})\Big] \cdot \text{cov}\Big[\hat{f}(\boldsymbol{a_n}), \hat{f}(\boldsymbol{Z_j})\Big]\, \text{q}(\boldsymbol{a_n})\, \text{d}\boldsymbol{a_n}$$

$$= \sum_{n=1}^{N} \hat{\sigma}_{ni}^2 \hat{\sigma}_{nj}^2 \sqrt{\frac{(\boldsymbol{\Sigma_a})_{nn}^{-1}}{\hat{\ell}_{ni} + \hat{\ell}_{nj} + (\boldsymbol{\Sigma_a})_{nn}^{-1}}} \cdot \exp\left(-\frac{1}{2}\frac{\hat{\ell}_{ni}\hat{\ell}_{nj}}{\hat{\ell}_{ni} + \hat{\ell}_{nj}}(\boldsymbol{Z_i} - \boldsymbol{Z_j})^2\right.$$

$$\left. -\frac{1}{2}\frac{(\boldsymbol{\Sigma_a})_{nn}^{-1}(\hat{\ell}_{ni} + \hat{\ell}_{nj})}{(\boldsymbol{\Sigma_a})_{nn}^{-1} + \hat{\ell}_{ni} + \hat{\ell}_{nj}} \cdot \left((\boldsymbol{\mu_a})_n - \frac{\hat{\ell}_{ni}\boldsymbol{Z_i} + \hat{\ell}_{nj}\boldsymbol{Z_j}}{\hat{\ell}_{ni} + \hat{\ell}_{nj}}\right)^2\right).$$

(25)

Note that the $\Psi$-statistics factorize along the data and we only need to consider the diagonal entries of $\boldsymbol{\Sigma_a}$. If all the data belong to the same output, the $\Psi$-statistics of the standard squared exponential kernel can be recovered as a special case. It is used to propagate the uncertainties through the output-specific warpings $\boldsymbol{g}$.

### A.4 Approximative Predictions

Using the variational lower bound in (5), our model can be fitted to data, resulting in appropriate choices of the kernel hyper parameters and variational parameters. Now assume we want to predict approximate function values $\boldsymbol{g_{d,\star}}$ for previously unseen points $\boldsymbol{X_{d,\star}}$ associated with output $d$, which are given by $\boldsymbol{g_{d,\star}} = g_d(f_d(a_d(\boldsymbol{X_{d,\star}})))$.

Because of the conditional independence assumptions in the model, other outputs $d' \neq d$ only have to be considered in the shared layer $\boldsymbol{f}$. In this shared layer, the belief about the different outputs and the shared information and is captured by the variational distribution $\text{q}(\boldsymbol{u_f})$. Given $\text{q}(\boldsymbol{u_f})$, the different outputs are conditionally independent of one another and thus, predictions for a single dimension in our model are equivalent to predictions in a single deep GP with nested variational compression as presented by Hensman and Lawrence [11].

## B  Joint models for wind experiment

In the following, we show plots with joint predictions for the models discussed in Section 4.2. Similar to Section 4.1, we trained a standard GP in Figure 6, a multi-output GP in Figure 7, a deep GP in Figure 8 and our model in Figure 9. All models were trained until convergence and multiple runs result in very similar models. For all models we used RBF kernels or dependent RBF kernels where applicable.

Each plot shows the data in gray and two mean predictions and uncertainty bands. The first violet uncertainty band is the result of the variational approximation of the respective model. The second green or blue posterior is obtained via sampling. For both the GP and MO-GP, we used the SVGP approximation [12] and since the models are shallow, the approximation is almost exact.

Figure 8 showcases the difficulty of training a deep GP model and the shortcomings of the nested variational compression. The violet variational approximation is used for training and approximates

the data comparatively well. As discussed above, the deep GP cannot share information, so the test sets cannot be predicted. However, as discussed in more detail in [12], the approximation tends to underestimate uncertainties when propagating them through the different layers and because of this, uncertainties obtained via sampling tend to vary considerably more. Because during model selection sample performance does not matter, the true posterior can be (and in this case is) considerably different.

Our approach in principle has the same problem as the deep GP. However, because of the strong interpretability of the different parts of the hierarchy, uncertainties within the model are never placed arbitrarily and because of this, the variational posteriors and true posteriors look much more similar. They tend to disagree in places when there is high uncertainty about the alignment.

**Figure 6:** GP

**Figure 7:** MO-GP

**Figure 8:** DGP

**Figure 9:** AMO-GP (Ours)