[Reviews · NeurIPS 2018]

Reviewer 1



This submission presents a "three-layer" Gaussian process for multiple time-series analysis: a layer for transforming the input, a layer for convolutional GP, and a layer for warping the outputs. This is a different "twist" or "favour" of the existing deep-GP model. Approximate inference is via the scalable version of variational inference using inducing points. The authors state that one main contribution is the "closed-form solution for the $\Phi$-statistics for the convolution kernel". Experiments on a real data set from two wind turbines demonstrates its effectiveness over three existing models in terms of test-log-likelihoods. [Quality] This is a quality work, with clear model, approximation and experimental results. In addition, Figure 3 has shown a illustrative comparison with existing models; results against these models are also given in Table 1. One short-coming is that the authors have not considered how their approach is better (perhaps in terms of inference) than a more straightforward model where the alignment is directly placed on the input without convolution. [Clarity] L41: I would describe the model as "nested" rather than "hierarchical", so as not to be confused with Bayesian hierarchical. Section 2: I think this entire section should be rewritten just in terms of time-series, that is, one-dimensional GP, and the bold-faced of $x$ and $z$ removed. This is because L69-L70 describe only a single output $a_{d}$ function, which means $z$ must be single dimension, and hence $x$ is only single dimension. If the multi-dimensional inputs are desired, then the paper has to perhaps use a multi-output function for $a_{d}$. Also, for equation 1, it has to be stated that the functions are applied point-wise. Since the authors cited both [14] and [19] for the "warped" part, I suggest that clearly state that the model follows the spirit of [14] rather than [19]. [Originality] I would dispute the claim on L145 that one "main contribution ... is the derivation of a closed-formed solution for the $\Phi$-statistics". This is because this is only for the RBF kernel, and it is simple a mathematically tedious step to get the expressions, given the previous work of [8] and [25, section B.1]. In addition, once the model is stated, the rest trivially fall into places based on existing work. [Significance] I commend the authors for proposing this model, which I think will be very useful for time-series analysis, and give yet another manner in which GP can be "nested"/"deepened". [25] M. K. Titsias and M. Lazaro-Gredilla. Variational Inference for Mahalanobis Distance Metrics in Gaussian Process Regression. NIPS, 26, 2013. [Comments on reply] The reply will be more convincing if it can in fact briefly address "how alignments can be generalized to higher dimensions".

Reviewer 2



This work presents a new Gaussian process model for multiple time series that contain a) different temporal alignments -or time warpings- (which are learned) and b) are related (in the multi-task sense), where the level of relation between tasks is also learned. This is applied to toy data and a very small real-world example of power generated by wind turbines. The model is well described and the diagrams used to explain it are useful to understand it. Relevant connections with existing literature (multi-ouput GPs, deep GPs, warped GPs, etc) are established. Connection with previous literature on GPs for time series alignment seems to be missing, for instance "Time series alignment with Gaussian processes", N Suematsu, A Hayashi - Pattern Recognition (ICPR), 2012. The model and the corresponding inference scheme, despite is laborious derivation, is by itself not very novel. It is a relatively straightforward combination of ideas from the above mentioned literature and requires the same tricks of introducing pseudo-inputs and variational inference to obtain a lower bound on the evidence. The main contribution could be providing the concrete derivation of the Phi-statistics for this particular model. I think the derivations are sound. However, I am more concerned about the practical applicability of this model: - The model is only applied to toy(ish) data. The computational cost is not discussed and might be very large for larger datasets, and with more than two signals. - The relation between tasks is governed by the set of hyperparameters \ell_{dd'}. In the simple examples given with only two tasks, this might work well. For cases with many signals, learning them might be subject to many local minima. - The learning of this model is very non-convex, since there are potentially multiple alignments and task-relatedness levels that achieve good performance (there are even multiple degenerate solutions for more that two outputs), so the hyperparameter optimization and inference can get very tricky for any non-trivial size dataset. Minor: - The clarity of the exposition could be improved. For instance, my understanding is that X is actually always one-dimensional and refers to time, but this isn't clear from the exposition of the paper. I incorrectly thought those would correspond to the locations of the turbines. - Continuing with the previous point, am I correct that the location of the turbines is not used at all? That would seem to provide valuable information about task relatedness. - There could be a comment mentioning that learning \ell_{dd'} is what learns the "task relatedness". After reading the authors' response: The response from the authors doesn't address the questions that I raised above. What's the computational cost? The authors manage to "address" this latter issue in no less than 11 lines of text without giving the computational complexity of the algorithm nor the runtimes. Also, they don't define X properly in the rebuttal either.

Reviewer 3



The authors propose an approach for non-linear time alignment of sequences in a shared latent space. More specifically they explain the observed time series data via a generative process: (i) first the inputs are passed through an alignment function, a, (ii) a shared latent process, f, is applied to the aligned signal, and, (iii) a warping function is responsible to introduce output specific attributes to the observe the final signal. In all three steps the authors impose Gaussian process (GP) priors to model the unknown functions, forming a set of deep GPs, one for each observed sequence. Additional structure is imposed on the function f by allowing it to model correlations between multiple outputs via a convolution with a number of independent processes. The key part of the paper is that the set of deep GPs is enforced to interact with each other via sharing information through the common middle layer, i.e., the function f is shared across the different Deep GPs. The model is learned via optimising the lower bound to the likelihood, which is obtained following a nested variational compression approach. Experimental results on toy and real world data demonstrate the advantages of the proposed model. The paper is nicely written and I really enjoyed the fact that it targets a specific problem/application from the very beginning. It may seem a bit weird at first, however, the reader really knows what is expected to follow and can definitely make the connection between the modelling choices and the targeted challenges. I really liked the idea of sharing the middle layer and the inducing variables u_{f,d} across the multiple sequences. However, the authors should be a bit more careful with the used notation. From my understanding *boldfaced* f is basically the collection of the function f evaluated on the data a(x). If this is correct then when the authors refer to a shared *boldfaced* f throughout the manuscript can be very misleading, especially when we talk about sequences with different cardinality. My biggest concern has to do with my understanding of the Psi statistics in Section 3.2 - First of all indices in this section are overloading the indices already defined. i is now used to denote the inducing point and n the point in the sequence, while i has been already used to denote the elements of X, i.e. X={x_i}_i=1:N. Can you please clean up this mess? - In both equations 6 and 7 we can see the Psi and Phi to be computed under some u. What is that u? Normally I would expect that the inducing input Z would be in the place of u. Have we assumed that the inducing inputs at the second layer are the inducing outputs of the previous layer? Then that would make sense, although, in that case we would need to integrate out also over u from the previous layer. In general I am very confused since in the same page in line 105 we have already defined different Z and u for f. Am I missing something? Please explain and correct me if I am wrong. The evaluation procedure is targeted to highlight the advantages of the proposed model, although the captions accompanying the figures do not always fully explain what is happening: - For example in Figure 3 there is no description about the continuous lines that are draws from the model; the reader has to wait till the next experiment and specifically Figure 5 in order to acquire that information. - Furthermore, in Figure 4 there is no description whatsoever of what the middle row (purple) line demonstrates. Is this the recovered latent function? In general I do not understand how is it possible to associate points in that plot with points in the two turbines since the two sets have different cardinalities. Why this is not the case in the toy experiment in Figure 3? Minor comments: - Figure 3 c & d, y labels for the plots in the third row should be g instead of y. Am I correct? - Same in figure 2 - References need update since you cite the arxiv version of many published papers. ---After rebuttal--- The authors have addressed my comments. I enjoyed the paper. I believe it can get improved by including an additional experiment with more than two signals to align or a more general non-linear warping function on real data. Since I have not suggested that in my original review I trust the authors would include such an experiment in the final version, if the paper goes through. I am still positive about the submission.